# Treatment Patterns and Pharmacoutilization in Patients Affected by Rheumatoid Arthritis in Italian Settings

**DOI:** 10.3390/ijerph18115679

**Published:** 2021-05-26

**Authors:** Valentina Perrone, Serena Losi, Veronica Rogai, Silvia Antonelli, Walid Fakhouri, Massimo Giovannitti, Elisa Giacomini, Diego Sangiorgi, Luca Degli Esposti

**Affiliations:** 1CliCon S.r.l., Health Economics and Outcomes Research, 40141 Bologna, Italy; elisa.giacomini@clicon.it (E.G.); diego.sangiorgi@clicon.it (D.S.); luca.degliesposti@clicon.it (L.D.E.); 2Eli Lilly Italy S.p.A., 50019 Sesto Fiorentino, Italy; losi_serena@lilly.com (S.L.); rogai_veronica@lilly.com (V.R.); antonelli_silvia@lilly.com (S.A.); 3Eli Lilly and Company Limited, Windlesham GU20 6PH, UK; fakhouri_walid@lilly.com; 4Eli Lilly Italy S.p.A., 00144 Roma, Italy; giovannitti_massimo@lilly.com

**Keywords:** biologic drugs, baricitinib, Janus kinase inhibitors, rheumatoid arthritis

## Abstract

This study aimed to evaluate the treatment patterns and pharmacoutilization of patients with rheumatoid arthritis (RA) in real-world settings in Italy. This retrospective observational analysis was based on administrative databases of selected Italian entities. All adult patients with RA diagnosis confirmed by ≥1 discharge diagnosis of RA (ICD-9-CM code = 714.0) or an active exemption code (006.714.0) were enrolled in 2019. Two cohorts were created: one included patients prescribed baricitinib, the other those prescribed biological disease-modifying antirheumatic drugs (bDMARDs). Overall, 47,711 RA patients were identified, most of them without DMARD prescription. As a first-line prescription, 43.2% of patients were prescribed conventional synthetic DMARDs (csDMARDs), 5.2% bDMARDs and 0.3% baricitinib. In 2019, 82.6% of csDMARD users continued with the same DMARD category, 15.9% had a bDMARD, while 1.5% had baricitinib as second-line therapy. Overall, 445 patients used baricitinib during 2019. During follow-up, baricitinib was prescribed as monotherapy to 31% of patients, as cotreatment with csDMARDs and corticosteroids to 27% of patients, with corticosteroids to 28% of patients and with csDMARDs to 14% of patients. In line with previous findings, a trend of bDMARD undertreatment was observed. The treatment patterns of baricitinib patients could help to better characterize patients eligible for new therapeutic options that will be available in the future.

## 1. Introduction

Rheumatoid arthritis (RA) is a debilitating progressive systemic inflammatory autoimmune disease. RA prevalence in Italy is estimated approximately at 0.41–0.48% [1,2]. RA is characterized by synovitis, and clinical features usually include arthralgia, polyarthritis and arthritis involving the small joints of the hands, feet and wrist; the features generally present in a symmetrical pattern, thus resulting in limitation of the range of motion as well as in impaired function of the affected joints [3,4]. In addition, RA patients often have extra-articular manifestations [5]. The etiology of this disease is not yet fully explained; it has been proposed that environmental factors could trigger the disease in subjects who are genetically predisposed [6].

Currently, the main therapeutic target for RA patients is clinical remission, with low disease activity as the best possible alternative [7]. A prompt start of therapy after diagnosis and tight monitoring of disease activity according to a “treat-to-target strategy” are important to obtain the best outcomes and to prevent functional impairment and disability [8,9].

Over the past 20 years, the management of RA has radically changed. The choice of treatments, which were previously mostly based on conventional synthetic disease-modifying antirheumatic drugs (csDMARDs), has expanded with the development of biological DMARDs (bDMARDs) and, more recently, with the new class of targeted synthetic DMARDs (tsDMARDs) [10]. International and national guidelines place this class at the same level as bDMARDs; both are recommended in case of an inadequate response to csDMARDs or when poor prognostic factors are present [7,11]. The tsDMARDs baricitinib and tofacitinib are orally administered Janus kinase (JAK) inhibitors currently licensed to treat adult patients affected by the moderate-to-severe active RA and who did not adequately respond to previous conventional therapies [12]. They can be administered as monotherapy or combination therapy. In Italy, baricitinib has been reimbursed by the National Health Service (NHS) since 2017, while tofacitinib was approved in 2018 [13,14].

The efficacy and safety of tsDMARDs have been thoroughly investigated in randomized clinical trials (RCTs) [12], which pose strict criteria that can limit the applicability of the results in real-life practice [15]. Real-world studies that analyze the pattern of use of new drugs in routine rheumatology practice consider unselected patients potentially representing the entire spectrum of disease severity, thus making it possible to account for the heterogeneity of the RA population with varying demographic and clinical characteristics [16].

We previously reported a description of treatment patterns and pharmacoutilization in RA patients during the first year after baricitinib reimbursement approval in Italy [17]. In the present study, we aim to provide an up-to-date scenario by evaluating the prescription patterns of RA patients, including baricitinib users, in 2019, in real-world settings of everyday clinical practice in Italy.

## 2. Materials and Methods

### 2.1. Data Sources

A retrospective observational analysis was carried out by integrating administrative databases from a pool of Italian entities from Veneto, Marche, Abruzzo, Apulia and Calabria Regions. These databases include the NHS provided healthcare services; therefore, they hold information meant to be used for reimbursement purposes. These healthcare administrative databases are large repositories of routinely collected data. Specifically, the main dataflows concern the following: the beneficiaries’ database, in which demographic characteristics are reported; pharmaceuticals databases, in which data are collected regarding the drugs dispensed by pharmacies reimbursed by NHS as the Anatomical Therapeutic Chemical (ATC) codes, the date and the number of drug dispensed; the hospitalization database, which contains all hospitalizations data for patients in the analysis, including the discharge diagnosis codes classified according to the International Classification of Diseases (Ninth Revision, Clinical Modification (ICD-9-CM)); and a payment exemption database, containing dates and codes of exemption—the latter allows the economic contribution for services/treatments for confirmed diagnosis to be avoided.

In full compliance with the European General Data Protection Regulation (GDPR) (2016/679), an anonymous univocal numeric code was attributed to each individual, thus allowing the electronic linkage of all records for each subject across the databases. All the results of the analyses were produced as aggregated. Informed consent was not required because obtaining informed consent was impossible for organizational reasons (pronouncement of the Data Privacy Guarantor Authority, General Authorisation for personal data treatment for scientific research purposes—n.9/2014). According to the Italian law regarding the conduct of observational analyses [18], the Ethics Committees of each participating entity were notified of this study and approved the study.

### 2.2. Study Population

All adult patients (≥18 years old) with an RA diagnosis in the calendar year 2019 (study period) were enrolled. The diagnosis was ascertained according to the criteria of at least one hospitalization with a discharge diagnosis (at primary or secondary level) with ICD-9-CM code 714.0 or the presence of an active exemption code (006.714.0), i.e., a payment waiver code given after confirmation of the diagnosis. Patients were classified according to the first prescription within the study period, then two cohorts were created: one included patients prescribed baricitinib (ATC code: L04AA37), and the other included those prescribed bDMARDs/tsDMARDs indicated for RA (abatacept (ATC: L04AA24), adalimumab (ATC: L04AB04), anakinra (ATC: L04AC03), certolizumab (ATC: L04AB05), etanercept (ATC: L04AB01), golimumab (ATC: L04AB06), infliximab (ATC: L04AB02), rituximab (ATC: L01XC02), sarilumab (ATC: L04AC14), tocilizumab (ATC: L04AC07), tofacitinib (ATC: L04AA29)). In the two cohorts, the index date (ID) was defined as the date of the first baricitinib or bDMARD prescription within the study period. Patients were then followed up from the ID to the end of data availability (31 December 2019).

In the baricitinib cohort, patients without a prescription for bDMARDs from 2010 (which marked the beginning of data availability within the database) to ID were defined as “bDMARD naïve”; others were regarded as “established”.

### 2.3. Study Variables

Demographic characteristics such as sex and age at inclusion were provided. In the baricitinib cohort, data on age and sex were collected at ID. Mean age at the time of the first record for RA (discharge diagnosis or exemption code) encountered from 2010 to ID was also reported as a proxy of age at RA diagnosis (which, however, could not necessarily correspond to age at onset). Treatment patterns were assessed in 2019 for all patients included. In the baricitinib and bDMARD cohorts, the treatments were also analyzed in the 12 months before the ID (pre-ID period) and during follow-up. The following therapies were considered: csDMARDs (azathioprine (ATC: L04AX01), ciclosporin (ATC: L04AD01), chloroquine (ATC: P01BA01), gold salts (ATC: M01CB01, M01CB03), hydroxychloroquine (ATC: P01BA02), leflunomide (ATC: L04AA13), methotrexate (ATC: L04AX03), sulfasalazine (ATC: A07EC01)), corticosteroids (ATC: H02), nonsteroidal anti-inflammatory drugs (NSAIDs) (ATC: M01), baricitinib, and the bDMARDs listed above.

The presence of comorbidities was measured by adapting the Charlson Comorbidity Index (CCI), which assigns a single score (minimum 0, maximum 6) to patients by weighting each concomitant disease identified in the 12 months prior to the ID (if none, the CCI score is 0). The comorbidities were identified from discharge diagnosis at primary and secondary levels. When a diagnosis was not available, the prescriptions of specific drugs were used as a proxy to determine the specific comorbidity. Switch was defined as a change of index therapy with a bDMARD during follow-up.

### 2.4. Statistical Analysis

All analyses were descriptive. Categorical variables are expressed as frequencies and percentages. Continuous variables are reported as means ± standard deviations (SDs). All analyses were performed using STATA SE version 12.0((StataCorp LLC, College Station, TX, USA).

## 3. Results

A total of 47,711 patients with RA diagnosis in 2019 were identified. The mean age was 67.9 years, and 26.9% of patients were male. As shown in Figure 1, as a first treatment encountered during the study period (calendar year 2019), more than two-fifths (43.2%) of patients were prescribed csDMARDs (mean age 65.1 years), 5.2% were given bDMARDs (mean age 57.3 years), and 0.3% were given baricitinib (mean age 60.4 years). However, the majority (51.3%) had no DMARD prescription (mean age 71.4 years); in this group, 39.4% were treated with symptom-related therapies such as NSAIDs or corticosteroids. Over 2019, the majority (82.6%) of csDMARD users (mean age 66.5 years) continued with the same category of DMARDs, while 15.9% had a bDMARD (mean age 58.3 years) and 1.5% had baricitinib (mean age 59.9 years) as a second-line therapy. Overall, the baricitinib cohort comprised 445 patients (Table 1): at index date, the mean ± SD age was 59.2 ± 12 years, 13.9% were male, and the CCI was negligible.

In the pre-ID period, 61% of baricitinib patients were treated with both csDMARD and corticosteroids, 20% were treated with corticosteroids, 13% were treated with csDMARD alone, and 6% did not receive those drugs (Figure 2a). The csDMARDs prescribed were mainly methotrexate (53.9%), leflunomide (18.4%) and hydroxychloroquine (13.5%) (Table 1). During a mean **±** SD follow-up of 5.4 **±** 2.3 months, the proportion of patients who received baricitinib as a monotherapy regimen (no csDMARDs or corticosteroids) was 31%, and that of cotreatment with both csDMARDs and corticosteroids was 27%, while 28% were cotreated with corticosteroids and 14% with csDMARDs (Figure 2b).

In the bDMARD cohort (*N* = 5767 patients), treatment patterns were distributed in the pre-ID period as follows: 37% of patients were prescribed csDMARD and corticosteroids, 22% were prescribed csDMARDs, 20% were prescribed corticosteroids and 21% did not have csDMARD/corticosteroid prescriptions (Figure 3a). During a mean **±** SD follow-up of 7.2 **±** 2.3 months (Figure 3b), 28% of patients received bDMARD monotherapy: specifically, patients were mainly treated with etanercept (10.2%), adalimumab (6.2%) and tocilizumab (3.5%). A total of 27% had concomitant csDMARD and corticosteroids, and 24% and 21% were cotreated with csDMARDs and corticosteroids, respectively (Figure 3b).

The following analyses focused on the baricitinib cohort. At index date, 63.6% (*N* = 283) of the patients were bDMARD naïve, while 19.3% and 17.1% were previously prescribed with 1 and 2 or more bDMARDs, respectively. The treatment patterns of bDMARD-naïve patients treated with baricitinib were similar to those observed in overall baricitinib users, as shown in Figure 4. Methotrexate was the most frequently prescribed csDMARD both before ID and during follow-up (60% and 30%, respectively, Table 2).

In the 12 months before the ID, the most frequently prescribed bDMARDs among established patients (*N* = 162) were etanercept (27.2%), tocilizumab (25.3%), adalimumab (13.6) and certolizumab (11.7), as shown in Table 3. Patients had a mean treatment duration of 0.5 years, and while 84.3% of them maintained the therapy, 13.3% and 2.5% switched once and two times, respectively, with a mean time to switch of 6.4 months. The bDMARDs more frequently prescribed after baricitinib were sarilumab (3.1%), etanercept (2.9%), abatacept (2.7%), golimumab (2.0%), adalimumab and certolizumab (0.9%).

## 4. Discussion

The treatment options for RA have rapidly evolved over the last few years, and new therapies targeting different mechanisms are currently emerging. Intending to evaluate the latest treatment landscape for RA in Italian clinical practice, we retrospectively assessed the treatment patterns among adult RA patients in a real-world setting in Italy. Since administrative databases contain data on healthcare resources reimbursed by the NHS, in line with previous studies, we focused our attention on the utilization of the most recent drugs reimbursed, which allowed us to collect a sufficient sample size (i.e., the tsDMARD baricitinib). The aim was to evaluate patients’ journey in terms of previous treatment, combination therapies, and utilization during follow-up to delineate a profile of patients that could be considered eligible for new treatment options for RA management. Our investigation offers an in-depth description of the therapeutic pathways of patients starting such therapies because of the availability of future therapeutic options for RA [12].

The demographic characteristics of the study population revealed that the mean age was generally higher in the conventional or symptomatic treatment groups than in the baricitinib and bDMARD groups. There are indeed several studies in the literature reporting that, in clinical practice, patients at a younger age are more likely to be prescribed bDMARD treatment than older patients [19,20,21]. Furthermore, compared to other drugs, fewer men were treated with baricitinib.

During the study period, a significant proportion of patients (51%) affected by RA were not treated with DMARDs and received no or only symptom-related therapies such as NSAIDs or corticosteroids. This result suggests a general underuse of DMARDs, in contrast with the Italian Society for Rheumatology (SIR) clinical practice guidelines and the European League Against Rheumatism (EULAR) guidelines, which both state that therapy with DMARDs should begin right after the diagnosis of RA is made [7,11]. While we cannot be certain of the reasons underlying the underuse, possible explanations could include contraindications to DMARD therapy or a low disease activity/progression that could lead patients to initially avoid DMARDs [22]. Similar evidence from the US shows that only 55.5% of patients initiated DMARD treatment within 1 year of diagnosis; specifically, 52.8% started csDMARDs, while 2.6% started b/tsDMARDs [23].

Most treated RA patients received csDMARDs. This could indicate a low disease activity state, a new diagnosis, or a trend of b/tsDMARD underprescription, which has been described in Italian contexts [24,25,26]. Regarding the general prescribing patterns, compared to our previous study [17], we observed a slight increase in the percentage of csDMARD users (from 39.4% to 43.2%), which corresponded to a mild decrease in DMARD-untreated patients (from 55.6% to 51.3%).

EULAR guidelines recommend and support combination therapy of bDMARDs and tsDMARDs with csDMARDs [7]. However, a monotherapy regimen in RA clinical practice was more common than expected, with approximately 30% of RA patients on bDMARD monotherapy, irrespective of which bDMARD was prescribed [27].In our study, the monotherapy regimen consistently accounted for 31% and 28% of baricitinib and bDMARD patients, respectively.

To the best of our knowledge, there are very few real-world studies on baricitinib utilization. Regarding patient characteristics, our population shares demographic similarities with our previous study and with another Italian study [17,28]. Differences in prescribing patterns were instead observed when comparing our results with other real-world studies on the same topic. In a Swiss study using data from the “Swiss Clinical Quality Management in Rheumatic Diseases” (SCQM) registry, baricitinib was prescribed to 20% of patients as a first-line therapy (csDMARD inadequate responders), 22% as a second-line therapy, 14% as a third-line therapy and 43% as fourth-line (or higher) therapy [29]. Our study, which used administrative databases, showed that the majority of patients who were prescribed baricitinib were bDMARD naïve. However, these differences should be interpreted with caution, given that Italy and Switzerland have different healthcare systems. However, to properly understand prescription by line of therapy, future research linking line of therapy to patients’ clinical outcomes is needed.

This study presents limitations. Our study population reflected clinical practice settings, and the results must be interpreted considering the observational nature of the study and the data source used. The main limitation is the absence of clinical information related to the state and the progression of RA disease, DAS28 value or other clinical outcome measures for disease activity, as well as other potential confounders that could have affected the results. Therefore, it was not possible to assess the reasons behind the choice of biological agents or baricitinib over conventional therapies, or the choices related to monotherapy or combination regimens. Similarly, the reason for undertreatment is not known and, in some cases, could potentially be due to patients without a clinical diagnosis of RA or related to prescribers’ decision based on the characteristics and the comorbidity profile of patients. The presence of side effects or intolerance to drugs prescribed is not retrievable. Ultimately, the results refer to the sample population in the analysis and may not be generalized to the overall population.

## 5. Conclusions

The study presented herein could be helpful to provide insights into the real management of RA patients in clinical practice after new therapeutic options enter the market. Compared to our previous findings, in the study population, we still observed a significant proportion of untreated patients. The treatment patterns observed could represent the heterogeneity of the study population; however, the low proportion of patients receiving b/tsDMARDs may reflect a tendency toward b/tsDMARDs undertreatment. The analysis performed on patients prescribed baricitinib could be useful to gain an understanding of the characteristics and therapeutic pathways of patients eligible for new therapeutic options that will be available in the future.

## Figures and Tables

**Figure 1 ijerph-18-05679-f001:**
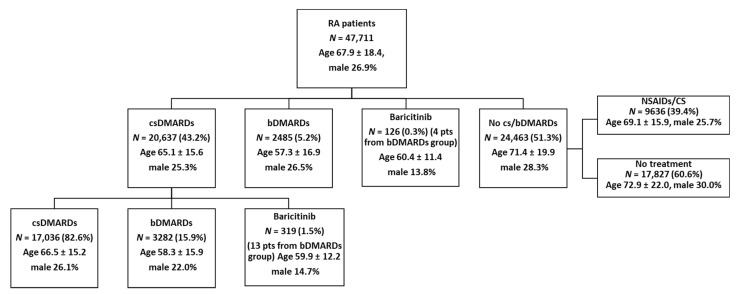
Treatment patterns of the included RA patients for the calendar year 2019. The first line refers to Table 2019, while the second line refers to treatments observed during follow-up. Abbreviations: RA, rheumatoid arthritis; csDMARDs or bDMARDs, conventional synthetic or biologic disease-modifying antirheumatic drugs; NSAIDs, nonsteroidal anti-inflammatory drugs; CS, corticosteroids.

**Figure 2 ijerph-18-05679-f002:**
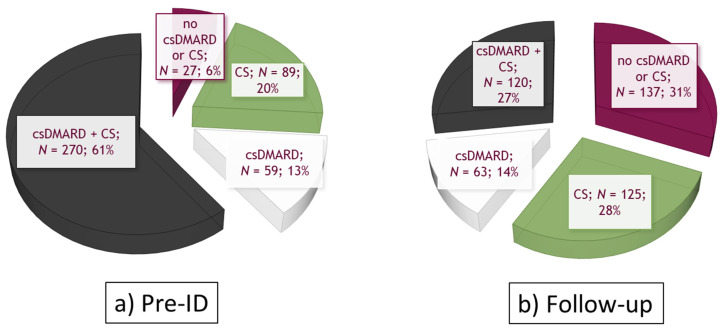
Treatment patterns in baricitinib cohort (*N* = 445) (**a**) in the 12 months before ID (pre-ID); (**b**) during follow-up. Abbreviations: csDMARDs, conventional synthetic disease-modifying antirheumatic drugs; CS, corticosteroids.

**Figure 3 ijerph-18-05679-f003:**
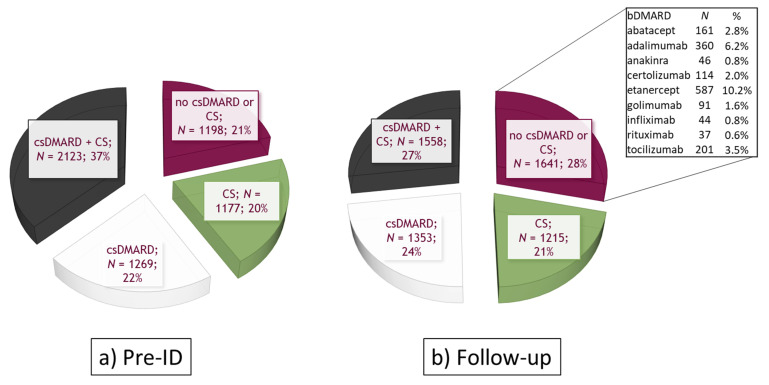
Treatment patterns in the bDMARD cohort (*N* = 5767) (**a**) in the 12 months before ID (pre-ID); (**b**) during follow-up. Abbreviations: csDMARDs, conventional synthetic disease-modifying antirheumatic drugs; CS, corticosteroids.

**Figure 4 ijerph-18-05679-f004:**
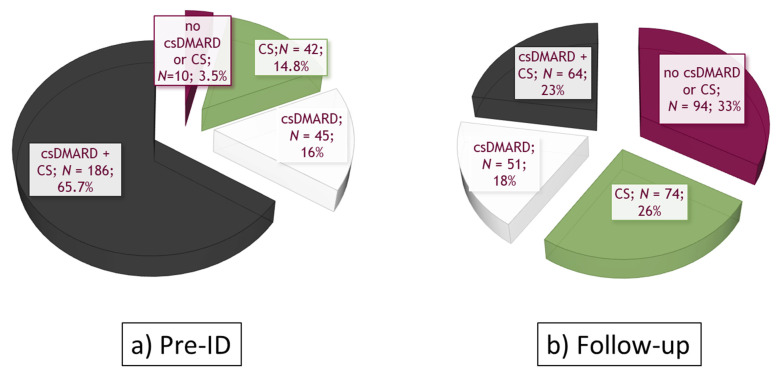
Treatment patterns in the bDMARD-naïve baricitinib users (*N* = 283) (**a**) in the 12 months before ID (pre-ID); (**b**) during follow-up. Abbreviations: csDMARDs, conventional synthetic disease-modifying antirheumatic drugs; CS, corticosteroids.

**Table 1 ijerph-18-05679-t001:** Clinical, demographic characteristics and previous use of csDMARD in the baricitinib cohort.

Patients treated with baricitinib, *n*	445
Age, mean ± SD	59.2 ± 12.0
Male, *n* (%)	62 (13.9)
Charlson Comorbidity Index, mean ± SD	0.1 ± 0.4
Age since RA diagnosis *, mean ± SD	51.3 (12.5)
Years since RA diagnosis * mean ± SD	8.2 ± 6.7
csDMARDs pre-ID	
Methotrexate	240 (53.9)
Leflunomide	82 (18.4)
Hydroxychloroquine	60 (13.5)
Sulfasalazine	21 (4.7)
Ciclosporin	5 (1.1)
Azathioprine	4 (0.9)
Others	4 (0.9)

Abbreviations: csDMARDs, conventional synthetic disease-modifying antirheumatic drugs; SD, standard deviation.* RA diagnosis was based on data availability of the databases (from 2010).

**Table 2 ijerph-18-05679-t002:** Most frequently prescribed csDMARDs to bDMARD-naïve patients (*N* = 283) during the pre-ID and follow-up period in the baricitinib cohort.

csDMARD	Pre-ID	Follow-Up
Methotrexate *N* (%)	169 (59.7)	84 (29.7)
Leflunomide *N* (%)	57 (20.1)	20 (7.1)
Hydroxychloroquine *N* (%)	50 (17.7)	11 (3.9)
Sulfasalazine *N* (%)	16 (5.7)	4 (1.4)

**Table 3 ijerph-18-05679-t003:** bDMARDs prescribed to established patients (*N* = 162) during the pre-ID period in the baricitinib cohort.

bDMARD Pre-Baricitinib	*N* (%)
Etanercept	44 (27.2)
Tocilizumab	41 (25.3)
Adalimumab	22 (13.6)
Certolizumab	19 (11.7)
Golimumab	14 (8.6)
Rituximab	7 (4.3)
Infliximab	6 (3.7)
Secukinumab	4 (2.5)

## Data Availability

All data used for the current study are available upon reasonable request next to CliCon s.r.l. which is the body entitled to data treatment and analysis by Local Health Units.

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
