# Peer review of "Treatment Patterns and Pharmacoutilization in Patients Affected by Rheumatoid Arthritis in Italian Settings"

_ijerph, 2021, doi:10.3390/ijerph18115679_

Round 1

Reviewer 1 Report

I have reviewed the author's response and revised manuscript that the authors submitted. I have no further comments.

Author Response

The Authors thank the Reviewer.

This manuscript is a resubmission of an earlier submission. The following is a list of the peer review reports and author responses from that submission.

Round 1

Reviewer 1 Report

The manuscript, ‘Treatment Patterns and Pharmacoutilization In Patients Affected by Rheumatoid Arthritis In Italian Settings: An Update 3 Of A Previous Real-World Analysis’ is a follow-up study, as mentioned in the previous study (https://doi.org/10.1007/s40744-020-00218-3). I have the following comments:

1. Unlike the paper title, this manuscript does not present any specific update or new treatment pattern compared with the previous study.

2.  In the studies using administrative claim data, RA patients are usually defined as patients who meet an algorithm based on an operational definition, such as having two or more RA diagnostic ICD codes plus DMARD prescription. Therefore, there is a possibility that patients who have not received DMARD for a year may not have RA. This may be an interesting fact, but ‘the pattern of how physicians are diagnosing RA’ would be beyond this journal's scope.

3. It is necessary to specify treatment patterns for biologic-naive baricitinib users during the pre-ID period, such as the list of csDMARDs (MTX, leflunomide, sulfasalazine...), and whether baricitinib was was treated as monotherapy or combined with a csDMARD.

Reviewer 2 Report

The paper of Perrone et al.  entitled Treatment patterns and pharmacoutilization in patients affected by RA in Italian setting presents statistical data on how patients with RA are really treated for Ra. The results form the study are similar to theses we are already aware. However the new data on baricitinib might be of special interest as we still accumulate data on real world clinical data  in baraitinib- treated patients. What is missing in my opinion is lack of any data on clinical characteristic, details why patients were switched to baricitinib or stopped JAKi. In spite of very large cohort , lack of these data makes the study incomplete. We know how many patients were treated and in such a way but we still do not know why. I am afraid that without these data study impact is reduced to statistical analysis of given regimens.

additional comments:

Please find more comments:
I hope it would be helpful for making  decision.
1.    If I understand well There is a discrepancy between fig 1 and  text  line 172-181. As we can seen at fig 1  baricitinib was used as the first line treatment in 126 pts but the text ( line 172) states it was used in 283 Again 36,4 % (19,3 and 17,1) were on bDMARDs – since fig suggests that all  previously treated patients were on csDMARDs

2.    Is it possible to specify which criteria for RA diagnosis was used as  more of 50% were not treated at all. Are they inactive( not needed to treat) or finally the other diagnosis was established. This also may bring suspicion that other than RA disease should be taken into account

3.    Why patients with  baricitinib as the first drug were chosen.  Do they differ from the others RA patients in term of age, disease duration disease activity , presence of poor prognosis factors etc. ( or maybe they have higher level of health insurance)The author should address this issues as data are of special importance making this paper interesting

4.    Some patients on baricitinib were switched to bDMARDs again it would be very interesting to readers why  they were switched-(eg side effects intolerance, cannot afford for such a treatment others?)  If patients failed BARI  what was the threshold to recognize drug ineffectiveness ( DAS28 value, patients outcome, patients preference, doctors’ s decision). Are patients who failed BARI differs from those who respond well to JAKi

5.    I understand that paper has many limitation , However it would be interesting to know whether any side effects occurred in BARI group, Maybe the Authors can collect them form databases. The JAKi treatment is still relatively new approach so any data on safety and effectiveness of such a treatment are of special importance

6.    It would be reasonable to put more data into table 1 including disease duration, age at disease onset, previous DMARD used, and failed disease activity at onset an follow up